# Role of Circular RNA in Brain Tumor Development

**DOI:** 10.3390/cells11142130

**Published:** 2022-07-06

**Authors:** Swalih P. Ahmed, Javier S. Castresana, Mehdi H. Shahi

**Affiliations:** 1Interdisciplinary Brain Research Centre, Faculty of Medicine, J. N. Medical College, Aligarh Muslim University, Aligarh 202002, India; swalihpahmed@gmail.com; 2Department of Biochemistry and Genetics, University of Navarra School of Sciences, 31008 Pamplona, Spain; jscastresana@unav.es

**Keywords:** circular RNA, miRNA, brain tumor, medulloblastoma, glioblastoma, pituitary adenoma, ependymoma, signaling pathway, diagnosis, treatment

## Abstract

Central nervous system tumors are a leading cause of cancer-related death in children and adults, with medulloblastoma (MB) and glioblastoma (GBM) being the most prevalent malignant brain tumors, respectively. Despite tremendous breakthroughs in neurosurgery, radiation, and chemotherapeutic techniques, cell heterogeneity and various genetic mutations impacting cell cycle control, cell proliferation, apoptosis, and cell invasion result in unwanted resistance to treatment approaches, with a 5-year survival rate of 70–80% for medulloblastoma, and the median survival time for patients with glioblastoma is only 15 months. Developing new medicines and utilizing combination medications may be viewed as excellent techniques for battling MB and GBM. Circular RNAs (circRNAs) can affect cancer-developing processes such as cell proliferation, cell apoptosis, invasion, and chemoresistance in this regard. As a result, several compounds have been introduced as prospective therapeutic targets in the fight against MB and GBM. The current study aims to elucidate the fundamental molecular and cellular mechanisms underlying the pathogenesis of GBM in conjunction with circRNAs. Several mechanisms were examined in detail, including PI3K/Akt/mTOR signaling, Wnt/-catenin signaling, angiogenic processes, and metastatic pathways, in order to provide a comprehensive knowledge of the involvement of circRNAs in the pathophysiology of MB and GBM.

## 1. Introduction

While central nervous system tumors account for a modest proportion of cancer diagnoses, they account for a significant proportion of cancer-related fatalities. Annually, almost 13,000 people die in the United States of America due to primary malignant brain and central nervous system tumors [1]. Medulloblastoma (MB) and glioblastoma (GBM) are the most prevalent CNS tumors in children and adults, respectively [2]. The WHO schemes are known for grading individual tumor classes (I, II, III, and IV) so that they can predict how they will behave. If there is no treatment, higher grade cancer (grades III and IV) is likely to be more aggressive than its lower grade counterpart in terms of how it looks and how quickly it gets better (grades I and II) [2]. A patient with the most common type of glioma, glioblastoma, can expect to live for only 15 months on average. As a result, radiation and chemotherapy have been more effective in treating childhood medulloblastoma, but the long-term side effects of these treatments can be very bad. The 5-year survival rate for this type of tumor has now reached 70% to 80% [3,4].

The most frequent pediatric brain tumor, medulloblastoma, has a 5-year survival rate of 71.9 percent. Medulloblastomas are the most common type of tumor on the cerebellum in childhood [5]. They are neuroepithelial tumors that account for 20% of all intracranial tumors in children, and for 40% of all childhood tumors in the fourth ventricle. Medulloblastoma is most common at 8 years old, but 30% of medulloblastomas happen in adults [6]. Medulloblastoma emerges from primitive neuroectoderm remnants in the fourth ventricle roof. It fills the ventricle and commonly invades the brainstem through the ependyma in the ventricle’s floor. Rarely, the tumor occurs in the cerebellar hemispheres [7]. The current standard of care for patients with an average risk is surgical resection followed by radiation and chemotherapy for medulloblastomas. However, radiation is frequently avoided in patients younger than three years of age due to the highly harmful effects on the developing brain [8]. High-risk patients with medulloblastoma are different from standard-risk patients because they are less than 3 years old, have metastases, or show residual tumor after surgery. They should receive more rigorous treatment than standard-risk patients [5].

The 5-year survival rate for the most common adult brain tumor, glioblastoma multiforme (GBM), is only 5.1% [9]. Glioblastoma is the most prevalent type of adult brain tumor [10]. But children are diagnosed with 8–9 percent of the illnesses [11]. Due to the tumor’s invasive, aggressive, and diffuse nature, treatment often consists of surgical excision followed by radiation and chemotherapy [12]. Malignant gliomas and medulloblastomas continue to provide significant treatment problems, many of which stem from the genetic and cellular heterogeneity of these tumor types. They include innate and acquired resistance, as well as the blood–brain barrier’s impediment to effective drug administration [2]. The difficulty of treating malignant tumors, along with the highly harmful effects of radiation on the developing brains of young children, makes alternate therapy extremely desirable.

Circular RNAs (circRNAs) are important among the non-coding RNA family members. They are single-stranded closed circRNA molecules that lack a 5’-end cap or a 3’-end poly(A) tail; and are formed through covalent bonding [13]. CircRNAs are abundant; they are stable structures; and they are extensively dispersed across a variety of tissues, cell types, and biological fluids, making them easily identifiable [14,15]. Numerous studies have established that circRNAs are differentially expressed in many types of tumor cells [16,17]. Additionally, researchers have discovered a substantial association between circRNA expression and several types of cancer, implying that circRNAs may operate as tumor inhibiting or tumor-promoting agents [18]. Numerous studies have been conducted to determine the function of various circRNAs in MB and GBM cell proliferation, migration, invasion, and apoptosis, and thus these molecules have been identified as viable therapeutic targets in the fight against MB and GBM [19,20,21,22,23]. Hence, a comprehensive understanding of circRNAs and their roles in the signaling and molecular mechanisms underlying MB and GBM may result in identifying more potent therapy methods. As a result, this review will focus on the underlying processes and signaling pathways impacted by circRNAs throughout the evolution of MB and GBM. The study’s findings may be useful for MB and GBM early diagnosis, pathological grading, targeted therapy, and prognostic evaluation.

## 2. Challenges, Perspective, and Clinical Significance

Medulloblastoma (MB), a type of primitive neuroectodermal tumor (PNET), is both the most prevalent malignant brain tumor in kids and the most common cause of cancer-related death in children [24,25]. The typical age of diagnosis for MB is six, with the majority diagnosed before the age of seventeen [24,26]. In the United States, there are about 500 cases of pediatric medulloblastoma every year [27,28]. These tumors make up 40% of all tumors in the posterior fossa. They can grow quickly and invade important structures, causing cerebellar dysfunction and causing cerebrospinal fluid to flow backwards. This pattern of growth can show up in the way people look and how they feel. Cerebellar signs are often seen in children, and they may have problems with coordination and walking. They may also experience headaches in the early morning, nausea, vomiting, papilledema, and double vision (hydrocephalus). The usual duration between symptom development and diagnosis is less than two to three months, but it can be even shorter [29].

Glioblastoma is the most prevalent malignant primary brain tumor, accounting for roughly 57% of all gliomas and 48% of all malignant primary central nervous system (CNS) tumors. Based on data from 2011 to 2015, the average yearly incidence of glioblastoma in the United States is 3.21 per 100,000 individuals [29]. The incidence varies according to age and sex. The median age of diagnosis is 65 years, with the highest incidence occurring in the 75–84 year age group. Males are 1.58 times more likely than females to acquire glioblastoma, with an annual age-adjusted incidence of 4.00 versus 2.53 per 100,000 population, respectively [30]. Glioblastoma prognosis remains poor. Advanced age, poor performance status, and insufficient resection extent are all well-established risk factors for poor outcomes. The median survival time for elderly individuals receiving only the best supportive care is four months [31,32]. The relative 1-year survival rate for patients diagnosed in the United States between 2000 and 2014 was 41.4 percent, up from 34.4 percent between 2000 and 2004, and 44.6 percent between 2005 and 2014. Despite these gradual increases in short-term survival rates over time, the 5-year survival rate has remained largely stable, at 5.8% five years after diagnosis [9,29,30].

The diagnosis of brain tumor necessitates the use of time-consuming diagnostic methods such as histological and molecular characterization, magnetic resonance imaging (MRI), and cytological examination of cerebrospinal fluid (CSF). As a result of the variability of clinical manifestations, which are typically characterized by easily ignored symptoms, it might take weeks or even months to confirm a diagnosis [33]. As a result, tumors are frequently rather big, and 20–30% of patients present with metastasis at diagnosis [34]. Treatment options for children with medulloblastomas remain uncertain. Over the last three decades, survival rates for children with medulloblastoma have increased, due to a variety of variables, including improved surgery, increased use of preoperative craniospinal radiation therapy, and, more recently, the addition of chemotherapy [35,36]. At the moment, conventional therapy for brain tumors is limited and consists of surgical excision of the tumor followed by radiation and adjuvant chemotherapy [37].

## 3. Circular RNA

Circular RNAs have been found to play critical roles in various biological processes, including cell proliferation, epithelial-mesenchymal transition, and tumor growth [38]. CircRNAs were discovered in eukaryotes more than two decades ago [39,40,41]. A vast group of mostly non-coding circular RNAs has important roles in tumorigenesis. Disease stage, outcome, age, and gender were associated with cirRNA patterns [42]. CircRNAs are a type of noncoding RNAs (ncRNAs) that have a continuous closed loop rather than 5′ caps and 3′ tails [21,43]. Numerous traits are conferred by the described features, including very high stability, which explains why they are so prevalent in the cytoplasm. CircRNAs are synthesized via back-splicing from pre-messenger RNA [21]. Certain circular transcripts may also be formed via direct RNA ligation, circularization of emancipated introns, or splicing of intermediates. Additionally, two different types of circular transcripts have been introduced: exon-skipping events, which happen in a small number of genes and involve internal splicing of skipped exons, and intron pairing-driven circularization, which happens when intronic motifs like Alu repeats cause the molecule to close by bringing the splice sites together [21,44]. CircRNAs were numerous and even substantially produced in mammalian cells compared to the host gene’s linear RNA isoforms [21]. CircRNAs have emerged as a diverse class of endogenous RNAs with primarily non-coding functions essential for development and illness [42]. However, increased interest in circRNA has emerged in recent years as a result of their increasing operations in human biology [45,46,47,48]. They have tissue-specific expression patterns and account for a sizable portion of cellular RNA, particularly in the brain [42]. CircRNAs have been implicated in miRNA response elements and gene expression regulation [21]. Circular RNAs have been involved in various biochemical processes (Figure 1).

## 4. Role of circRNAs in Medulloblastoma

Medulloblastoma (MB), a primitive neuroectodermal tumor (PNET), is the most prevalent malignant brain tumor in children and the major cause of childhood cancer-related mortality [9,27,28,29,49], it is a prevalent malignant pediatric brain cancer and is the second leading cause of child death after leukemia [50]. MB is frequently found in the cerebellum; the primary symptoms and signs are produced by intracranial hypertension and hydrocephaly [51]. MB is currently treated mostly through surgery, radiation, and chemotherapy. While MB is radiation and chemotherapy sensitive, infection, peripheral neuropathy, ototoxicity, and myelosuppression are common secondary side effects of excessive treatment in children who are often in the formative stage [51]. Patients at average risk (defined as children older than three years of age who have had their tumor nearly completely removed and are free of metastatic illness) have an estimated 5-year overall survival rate of 85% [52,53,54]. Patients with high-risk malignancies (tumors that develop in children younger than three years of age with less than complete resection and metastatic disease at presentation) have a survival rate of closer to 60–70 percent [54,55]. CircRNAs have been shown to modulate cancer-related processes including tumorigenesis, tumor development, and apoptosis [16,17].

Additionally, researchers have discovered a substantial association between circRNA expression and several types of cancer, implying that circRNAs may operate as tumor-inhibiting or tumor-promoting agents [18]. Numerous studies have been conducted in this context to determine the role of various circRNAs in MB cell proliferation, migration, invasion, and apoptosis. Therefore, these compounds have been presented as feasible therapeutic targets in the fight against MB. Thus, a better understanding of circRNAs and their roles in the signaling and molecular pathways underlying MB may result in the identification of more effective therapy methods. Most circRNAs were downregulated in MB tissues, consistent with prior findings in breast cancer, hepatocellular carcinoma, laryngeal cancer, colorectal cancer, gastric cancer, and prostate adenocarcinoma [21]. However, the role of circRNAs in medulloblastoma (MB) remains uncertain [21]. The purpose of this work was to determine the expression profiles of circRNAs that regulate tumor cell proliferation and growth in MB.

The levels of expression of eight distinct circ-DTL (hsa-circ-0000179), circRNAs [circ- SKA3 (hsa-circ-0029696), circ-CRTAM, circ- RIMS1-1 (hsa-circ-0132250), circ-MAP3K5 (hsa-circ-0006856), circ-RIMS1-2 (hsa-circ- 0076967)] circ-SKA3, and circ-DTL were tested. Circ-SKA3 and circ-DTL using short interfering RNAs and overexpressed their host genes to study their function in the pathogenesis of MB. 33 circRNAs showed differential expression in MB tissues, three of which were upregulated and thirty of which were downregulated; six of these circRNAs were successfully experimentally confirmed. By modulating the expression of host genes, upregulated circ-SKA3 and circ-DTL enhanced proliferation, migration, and invasion in vitro. This innovative work used circRNA profiling to indicate that circ-SKA3 and circ-DTL were critical in the carcinogenesis and progression of MB and may be viewed as novel and prospective biomarkers for diagnosis and new targets for treatments [21]. This study shows that circ-SKA3, circ-DTL, and circ-CASC15 are upregulated MB. Circular RNAs circ-UNC13C, circ-BRWD3, circ-CNTN6, circ-CRTAM, circ-MCU, circ-RIMS1-1, circ-FLT31, circ-DGKH, circ-FLT3-2, circ-SPHKAP, circ-GRM1, circ-GABRB2, circ-RIMS1-2, circ-ICA1, circ-GRIK2, circ-ATP8A2, circ-EPHX2, circ- WAC, circ-TENM1, circ-SNORD109A, circ-UNC13C, circ-GRIK2, circ-MAP3K5, circ-CAMKK2, circ-SVEP1, circ-CADPS2, circ-CAMK4-1, and circ-CAMK4-2 inhibited medulloblastoma growth [21] (Figure 2 and Table 1).

Circ-SKA3 and FOXM1 levels were raised, whereas miR-383-5p levels were decreased in MB tissues. Circ-SKA3 was shown to sponge miR-383-5p, which targets FOXM1. In vitro, suppressing circ-SKA3 slowed cell proliferation, migration, and invasion while inhibiting xenograft tumor growth in vivo. miR-383-5p reduced MB cell proliferation, migration, and invasion while increasing apoptosis via FOXM1. Circ-SKA3 [15] was substantially expressed in MB, and circ-SKA3 was found to be involved in regulating cell proliferation, migration, and invasion in MB [17]. With respect to circ-SKA3 expression in MB specimens and cells, functional investigations demonstrated that knocking down circ-SKA3 decreased not only MB cell proliferation, migration, and invasion in vitro, but also tumor initiation and growth in vivo, it identified miR-326 as a target of circ-SKA3 and demonstrated that it might also interacts with ID3. Circ-SKA3 increased the development of MB in vitro and in vivo by boosting ID3 expression via miR-326 targeting, which may be another technique for treating MB in future [20] (Figure 2 and Table 1).

## 5. Role of circRNAs in Glioblastoma

Glioblastoma multiforme (GBM) is still considered a lethal form of brain cancer, accounting for around half of all initial brain tumors [56]. It is an astrocytoma of WHO grade IV that accounts for around 30% of all brain cancers [57]. This is an extremely vascularized and infiltrating tumor [58]. GBM has remained as an incurable cancer with an average survival time of approximately 12–15 months [56,59]. It is a glial cell tumor with a poor prognosis and a high mortality rate in adults [60]. GBM is resistant to apoptosis induction and overexpresses anti-apoptotic proteins [61]. The propensity to proliferate rapidly has been identified as a characteristic of GB, resulting in therapies followed by a poor clinical evolution [62]. The WHO’s current international standard for naming and diagnosing glioma classifies these tumors into four subtypes. More broadly, gliomas are classified as low-grade gliomas (LGG, WHO I, and II) or high-grade gliomas (HGG, WHO III, and IV). Grade I gliomas are benign lesions with little proliferative potential to be healed surgically. Still, grades II to IV gliomas are highly infiltrative tumors and constitute the most frequent and malignant glioma grades [63]. Numerous studies have been conducted in this context to determine the function of various circRNAs in GBM cell proliferation, migration, invasion, and apoptosis. Therefore these molecules have been introduced as viable therapeutic targets in the fight against GBM [64,65,66]. It has emerged that circRNAs are key oncogenic drivers and tumor suppressors in glioblastoma and glioma of adults [42]. There may be better ways to treat GBM if we know more about how circRNAs work and how they affect GBM-related signaling and molecular mechanisms. So, this study will look into how circRNAs affect the basic mechanisms and signaling pathways that happen during the growth and spread of GBM.

### 5.1. CircRNAs Responsible for the Up-Regulation of Glioblastoma

Hsa-circ- 0046701 was highly elevated in glioma tissues and cell lines, and knockdown of hsa-circ-0046701 decreased cell proliferation and invasion in glioma tissues and cell lines. Luciferase reporter experiments revealed that hsa-circ-0046701 acts as a sponge for miR-142-3p and is involved in regulating ITGB8 transcriptional activity. Silencing of hsa-circ-0046701 could result in an increase in miR-142-3p expression, which in turn resulted in a decrease in ITGB8 expression. hsa- circ- 0046701/miR-142-3p/ITGB8 axis may play essential regulatory functions in the pathogenesis and progression of glioma [67] (Figure 3 and Table 2).

The expression of circBAP2 was increased in glioblastoma. In vitro, circ-UBAP2 upr regulates cell proliferation, migration, and invasion while decreasing apoptosis and regulating tumor development in vivo. Circ-UBAP2 was found to be directly linked to miR-1205 and miR-382. miR-1205 and miR-382 were found to be involved in the regulation of circ-UBAP2 silencing on glioma cell behavior. Additionally, miR-1205 and miR-382 were functional targets of GPRC5A in modulating glioma cell activities. Moreover, cirBAP2 induced GPRC5A expression in glioma cells via miR-1205 or miR-382. Circ-UBAP2 knockdown inhibited malignant glioma development in part by downregulating GPRC5A via miR-1205 and miR-382. UBAP2 specifically targeted miR-1205 and miR-382, which had been identified as tumor suppressors in glioma [68] (Figure 3 and Table 2).

Circ-PARP4 up-regulates glioma cell proliferation, migration, invasion, and epithelial-mesenchymal transformation. In vivo and in vitro investigations indicated that circ-PARP4, as a miRNA sponge, directly interacted with miR-125a-5p, which then controlled FUT4 to produce the oncogenic influence on glioma behavior. These findings highlight the activities of circ-PARP4 in influencing glioma progression through the miR-125a-5p/FUT4 pathway. miR-125a-5p has been reported to inhibit glioma cell proliferation and to promote cell differentiation by targeting TAZ. Suppression of miR-125a-5p can restore malignant phenotypes after inhibiting oncogene BCYRN1 in glioma. Down-regulated miR-125a-3p has been seen in CD133+ stem-like GBM cells compared with the CD133+ cells, and it can induce the differentiation of stem-like GBM cells, suggesting its involvement in the regulation of glioma stem cells. miR-125a-3p is considered as a tumor suppressor underlying the regulation of oncogenic circ-PARP4, which provides a novel and potential target for glioma therapy [69] (Figure 3 and Table 2).

Hsa-circ-0008344 silencing showed inhibited glioblastoma cell growth, colony formation, migration, and invasion but enhanced cell apoptosis in in vitro experiments. Bioinformatics predicts that hsa-circ-0008344 may interact with numerous miRNAs, including miR-433-3p and miR-450b-3p, whereas miR-433-3p has been shown to decrease cell proliferation and boost chemosensitivity in glioblastoma by targeting CREB, it is hypothesized that hsa- circ- 0008344 can sponge tumor suppressor miRNAs, resulting in the disinhibition of the expression of particular targeted oncogenic genes, which leads to the progression of glioblastoma. Hsa-circ-0008344 expressions are increased in glioblastoma and may have a role in the progression of this cancer [65] (Figure 3 and Table 2).

Hsa-circ-0012129 expression was dramatically enhanced in cell lines and glioma tissues; knockdown of hsa-circ-0012129 greatly inhibited the proliferation, migration, and invasion capacities of cells U373 and SHG44. A dual-luciferase reporter assay revealed that hsa-circ-0012129 shared a complementary binding site and that its expression had a negative effect on miR-661. MiR-661 rescue studies demonstrated that it could reverse the effects of hsa-circ-0012129 on glioma cell survival, migration, and invasion in vitro. This study’s findings suggested that circRNA hsa-circ-0012129 may operate as a natural miR-661 sponge in human glioma cells and that miR-661 may have inhibitory effects on circ-0012129 expression. Hsa-circ-0012129 has the potential to serve as a diagnostic or predictive biomarker for human glioma, as well as a therapeutic target [70] (Figure 3 and Table 2).

Circ-CDC45 and CSF-1 expression was increased in GBM tissues and cells; however, miR-485-5p expression was decreased. In vivo, silencing circ-CDC45 or CSF-1 inhibited GBM cell proliferation, invasion, migration, and tumor development. Circ-CDC45 favorably regulated CSF-1 expression by targeting miR-485-5p. miR-485-5p inhibition reduced the biological effects of circ-CDC45 deregulation in GBM cells [71] (Figure 4 and Table 2).

Circ-0029426 overexpression was associated with a poor prognosis in GBM, and circ- 0029426 restorations increased GBM cell proliferation while inhibiting cell apoptosis. The expression of Circ-0029426 is highly associated with the clinical severity and prognosis of patients. Circ-0029426 significantly increased cell proliferation, migration, and invasion while inhibiting cell death. Circ-0029426 was predicted and confirmed to sponge miR-197. More critically, circ-0029426’s carcinogenic properties are partially related to its inhibition of miR-197. Although the chemical mechanism by which circ-0029426 works has not been extensively investigated, circ-0029426 can still be considered a possible treatment target for GBM [72] (Figure 4 and Table 2).

Five cir-cRNAs that are overexpressed are found, including circ-ENTPD7 (hsa-circ-0019421), hsa-circ-0040705, hsa-circ-0003026, hsa-circ-0040719, and hsa-circ-0040708. Five more circRNAs were discovered as underexpressed, including hsa-circ-0000722, hsa-circ-0040723, hsa-circ-0040733, hsa-circ-0040738, and hsa-circ-0007361 [73] (Figure 4 and Table 2).

In glioblastoma tissues, circ-ENTPD7 (hsa-circ-0019421) expression was increased. When circ-ENTPD7 expression levels were elevated in glioblastoma patients, Kaplan–Meier analysis revealed a low overall survival. Circ-ENTPD7 silencing decreased glioblastoma cell motility and growth. Additionally, circ-ENTPD7 worked as a sponge for miR-101-3p, regulating ROS1 expression and promoting glioblastoma cell proliferation and motility [73] (Figure 4 and Table 2).

High levels of circ-FLNA expression were linked to a poor outcome in GBM. miR-1993p later suggested to be a circ-FLNA target. Following circ-FLNA knockdown, the inhibition of cell growth and invasion by miR-1993p was partially reversed. Finally, the current study’s findings discovered unexpected roles for circ-FLNA in GBM and suggested that the circ-FLNA/miR-1993p signaling axis may play a crucial role in GBM progression. As a result, circ-FLNA could be a unique target for the diagnosis and treatment of GBM [38] (Figure 4 and Table 2).

In vitro and in vivo hsa-circ-0076248 downregulation or miR-181a upregulation could inhibit glioma growth and invasion and significantly increase temozolomide chemotherapy sensitivity. Downregulating hsa-circ-0076248 or upregulating miR-181a could boost p53 and SIRT1 expression. The downregulation of SIRT1 expression appears to be the mechanism by which hsa-circ-0076248 governs glioma development and invasion. This implies that hsa-circ-0076248, miR-181a, and SIRT1 may be viable therapeutic feasible targets for glioma [74] (Figure 4 and Table 2).

The most prevalent circRNA generated from LGMN was hsa-circ-0033009 (circ-LGMN). Circ-LGMN expression was elevated in high-grade glioma (HGG), and it was associated with a worse outcome in patients with glioma. Circ-LGMN overexpression boosted the proliferation and invasion of GBM cells. Circ-LGMN acts as a sponge for miR-127-3p, preventing the degradation of LGMN mRNA by miR-127-3p, resulting in enhanced LGMN protein expression. Treatment of GBM cells overexpressing circ-LGMN with a miR-127-3p mimic inhibited proliferation and reduced invasion. Furthermore, overexpression of circ-LGMN enhanced GBM malignancy in vivo, whereas overexpression of miR-127-3p reversed this effect. Taken together, circ-LGMN is a new tumor-promoting circRNA that promotes LGMN expression by sponging miR-127-3p. Thus, targeting the circ-LGMN/miR-127-3p/LGMN axis for GBM treatment may be a potential therapeutic option [75] (Figure 4 and Table 2).

Circ-FOXO3 expression was considerably increased in GBM tissues compared to normal tissue. Proliferation and invasion of GBM cells were decreased when circ-FOXO3 was knocked down and increased when circ-FOXO3 was overexpressed. Further biochemical investigation revealed that circ-FOXO3 acted as a competitive endogenous RNA (ceRNA) to promote nuclear factor of activated T cells 5 (NFAT5) production by sponging both miR-138-5p and miR-432-5p. Notably, miR-138-5p/miR-432-5p inhibitors were able to reverse tumor inhibition caused by circ-FOXO3 downregulation in GBM cells. Additionally, GBM cells that expressed less circ-FOXO3 generated less aggressive tumors in vivo. Circ-FOXO3 can act as a regulator in GBM, and microRNA sequestration caused by ceRNA may be a therapeutic option for GBM [76] (Figure 4 and Table 2).

Circ-0074027 contributes to the development of GBM through modulating miR-518a-5p/IL17RD signaling. NEAT1 promotes colorectal cancer carcinogenesis by sponging miR-193a-3p, hence increasing IL17RD expression. The current study established that elevated expression of IL17RD may accelerate cell progression in GBM. Additionally, circ-0074027 acts as a miR-518a-5p sponge, increasing IL17RD expression and conferring carcinogenic characteristics. Thus, the circ-0074027/miR-518a-5p/IL17RD network enables a novel element of GBM treatment [66] (Figure 4 and Table 2).

Circ-SKA3 expression is increased in GBM and associates with poor prognosis. Circ-SKA3 may inhibit miR-1 expression by methylation, hence promoting GBM cell growth. Circ-SKA3 expression was increased in GBM tissues and was inversely linked with miR-1. High levels of circ-SKA3 and low levels of miR-1 were shown to be substantially associated with poor survival in GBM patients. In GBM cells, overexpression of circ-SKA3 boosted miR-1 gene methylation and decreased miR-1 expression. The CCK-8 experiment demonstrated that overexpression of circ-SKA3 decreased miR-1’s inhibitory effect on cell growth. Thus, circ-SKA3 may suppress miR-1 expression in GBM via methylation, thereby promoting cancer cell growth [77] (Figure 4 and Table 2).

Circ-0001801 helped GBM cells grow, move, invade, and become more like other cells by interacting with miR-628-5p. Circ-0001801 knockdown inhibited cell proliferation, migration, invasion, and epithelial-mesenchymal transition (EMT). Then, using the Dual-Luciferase reporter assay, the interaction between miR-628-5p and circ-0001801 or HMGB3 was confirmed. Reduced miR-628-5p expression in GBM tumors and cells established miR-628-inhibitory 5p’s activity. Additionally, inhibition of miR-628-5p may be able to reverse the suppressive effect of circ-0001801 silencing on GBM cell growth and EMT. Increased expression of HMGB3 may compensate for the inhibitory effect of circ-0001801 silencing on GBM cell growth. Circ-0001801 promoted cell survival, migration, invasion, and EMT in GBM via absorbing miR-628-5p. This study identifies novel biomarkers for the diagnosis and treatment of GBM [78] (Figure 4 and Table 2).

Silencing circ-FOXM1 inhibited the growth of GBM cells, as well as the growth of the tumor. miR-577 may be sponged by circ-FOXM1, and inhibiting it may reverse the inhibitory effect of circ-FOXM1 deregulation on GBM progression. E2F5 was a miR-577 target, and its silencing had the same effect on GBM development as circ-FOXM1 silencing. Circ-FOXM1 controlled E2F5 expression positively, whereas miR-577 regulated E2F5 expression negatively. In conclusion, this research established that might act as a sponge, thereby accelerating the progression of GBM by targeting E2F5, implying that circ-FOXM1 could be used as a biomarker for GBM treatment [79] (Figure 4 and Table 2).

Circ-NF1 expression was found to be increased in GBM and was found to be associated with patient survival. Circ-NF1 siRNA knockdown increased mature miR-340 expression in GBM cells, but not the precursor miR-340. Cell proliferation assays revealed that suppressing circ-NF1 siRNA and overexpression of miR-340 inhibited GBM cell proliferation. Additionally, the miR-340a inhibitor inhibited the proliferation-promoting effect of circ-NF1 siRNA silencing. In GBM cells, circ-NF1 inhibited miR-340 maturation and lowered its expression level. These data show that circ-NF1 may be a good candidate for GBM therapy [80] (Figure 4 and Table 2).

Circ-MELK is a sponge for the tumor-suppressing miR-593, specifically targeting the oncogenic gene Eph receptor B2 (EphB2). To assess the interactions between miR-593 and circ-MELK or EphB2, dual-luciferase reporter assays were used. Circ-MELK expression was increased in GBM, acting as an oncogene and regulating GBM mesenchymal transition and GSC maintenance through miR-593 sponging. Additionally, it discovered that EphB2 was implicated in GBM carcinogenesis produced by the circ-MELK/miR-593 axis. This function creates a window of opportunity to make a potential therapeutic target for gliomas [81] (Figure 4 and Table 2).

Circ-NUP98 and pre-mature miR-519a-3p expression were increased in GB; however, mature miR-519a-3p expression was decreased. Circ-NUP98 was negatively connected with mature miR-519a-3p but favorably correlated with pre-mature miR-519a-3p across cancer tissues. Circ-NUP98 was found in both the nucleus and cytoplasm of GBM cells and interacted with pre-mature miR-519a-3p. Circ-NUP98 boosted the expression of pre-mature miR-519a-3p in GBM cells while decreasing the expression of mature miR-519a-3p. BrdU and cholecystokinin octapeptide (CCK-8) tests demonstrated that overexpression of circ-NUP98 inhibited miR-519a-3p-mediated cell proliferation inhibition. Circ-NUP98 increased tumor size, which resulted in dramatically decreased mouse survival. Circ-NUP98 inhibits miR-519a-3p maturation, hence promoting GBM cell proliferation [82] (Figure 4 and Table 2).

#### 5.1.1. Mapks Pathway

Circ-PITX1 could act as a molecular sponge to make miR-379–5p less effective. miR-379–5p also targeted MAP3K2 as an upstream target. Moreover, miR-379–5p, MAP3K2, and circ-PITX1 were present in the RISC complex simultaneously. In addition, miR-379–5p could enhance the expression of MAP3K2 through circ-PITX1. Thus, circular RNA PITX1 acts as a ceRNA to regulate the miR-379–5p/MAP3K2 axis and promotes glioblastoma growth. Circ-PITX1 may be a novel research focus in GBM [83] (Figure 3; Table 2).

Circ-PITX1 expression was substantially higher in GBM tissues compared to control tissues. Additionally, this research established that circ-PITX1 aided in the formation of GBM by acting as a competitive endogenous RNA that absorbed miR-584-5p, thereby regulating KPNB1 expression. The roles of circ-PITX1 proposed a mechanism for circ-PITX1 involvement in the progression of GBM. In GBM tissues, circ-PITX1 expression was elevated and was inversely linked with miR-584-5p expression. Additionally, it was established that circ-PITX1 induced cancer in GBM through targeting the miR-584-5p/KPNB1 axis, implying that circ-PITX1/miR-584-5p/KPNB1 could be used as a diagnostic marker for GBM patients [84] (Figure 3 and Table 2).

Circ-ASAP1 expression was shown to be considerably higher in recurrent GBM tissues and TMZ-resistant cell lines. Circ-ASAP1 overexpression increased GBM cell proliferation and TMZ resistance, in which circ-ASAP1 knockdown reduced. Further research demonstrated that circ-ASAP1 boosted NRAS expression by sponging miR-502-5p. Furthermore, circ-ASAP1 depletion restored the sensitivity of TMZ-resistant xenografts [85] (Figure 3 and Table 2).

Circ-ASAP1 expression was very high in recurrent GBM tissues and TMZ-resistant cell lines. Circ-ASAP1 overexpression led to more GBM cell growth and resistance to TMZ, which could be reduced by knocking down circ-ASAP1. Further tests showed that circ-ASAP1 made NRAS more active by sponging miR-502-5p. In addition, circ-ASAP1 depletion restored the sensitivity of TMZ-resistant xenografts to TMZ treatment in the clinic. Circ-ASAP1 has then regulatory roles in GBM, and competing endogenous RNA (ceRNA)-mediated microRNA sequestration might be a good way to treat GBM [85] (Figure 3 and Table 2).

Circ-MAPK4 served as an oncogene in gliomas, was inversely regulated, and was associated with the clinico-pathological stage of gliomas (*p* < 0.05). Following that, circ-MAPK4 increased glioma cell survival and prevented apoptosis in vitro and in vivo. Additionally, circ-MAPK4 regulated the p38/MAPK pathway, which influenced the development and apoptosis of gliomas. miR-125a-3p, exhibited tumor-suppressive action by modulating the p38/MAPK pathway, which was elevated when circ-MAPK4 was inhibited and could be brought down by circ-MAPK4. Inhibition of miR-125a-3p partially compensates for the increased phosphorylation of p38/MAPK and the increased quantity of apoptosis-inducing protein caused by circ-MAPK4 knockdown. Circ-MAPK4 has a vital role in glioma cell survival and apoptosis by modulation of miR-125a-3p, which may provide a novel therapeutic target for the treatment of gliomas [86] (Figure 3 and Table 2).

Circular TTBK2 expression was increased in glioma tissues and cell lines, whereas linear TTBK2 expression was not altered in glioma tissues or cells. Increased circ-TTBK2 expression facilitated cell proliferation, migration, and invasion while inhibiting apoptosis. miR-217 expression levels were decreased in glioma tissues and cell lines. Additionally, It discovered that circ-TTBK2, but not linear TTBK2, behaved as a sequence-specific miR-217 sponge. Furthermore, increased circ-TTBK2 expression lowered miR-217 expression, resulting in a reciprocal negative feedback loop in an Argonaute2-dependent manner. Additionally, restoration of miR-217 dramatically reversed the stimulation of glioma growth by circ-TTBK2. HNF1 was a direct miR-217 target and acted as an oncogene in glioma cells. Surprisingly, the combination of circ-TTBK2 silencing and miR-217 overexpression resulted in tumor regression in vivo. Inhibition of the circ-TTBK2/miR-217/HNF1/Derlin-1 axis may be a viable target for human gliomas [87] (Figure 3 and Table 2).

#### 5.1.2. PI3K/AKT Pathway

Circ-0000215 and CXCR2 expression were significantly increased in glioma cells and tissues. Circ-0000215 overexpression boosted proliferation, invasion, and EMT while prevented apoptosis in glioma cells, whereas circ-0000215 knockdown had the opposite impact. Additionally, miR-495-3p, a sponge RNA derived from circ-0000215, suppressed glioma cell proliferation, invasion, and EMT. CXCR2 was a particular target of miR-495-3p, which negatively affected the CXCR2/PI3K/Akt pathway. However, the effects of miR-495-3p were all diminished when circ-0000215 was overexpressed. Circ-0000215 acts as a competitive endogenous RNA by sponging miR-495-3p, increasing glioma growth via the CXCR2 axis. By targeting the miR-495-3p/CXCR2/PI3K/Akt axis, overexpression of circ-0000215 can increase glioma growth. It is beneficial for the early detection and treatment of glioma, as well as for assessing its stage and prognosis [88] (Figure 3 and Table 2).

Circ-0037655 was more common in glioma tissues and cell lines (U251 and SHG-44) than in normal tissues and cell lines. Inhibiting circ-0037655 could make glioma cells less able to live and spread. Circ-0037655 is a sponge for miR-214, and when miR-214 is blocked, si-circ-0037655 can make cells less able to live and invade. Over-producing miR-214 could make p-Akt less likely to be found in the body (PI3K pathway indicator). This study looked at the expression of circRNAs in gliomas. It found that circ-0037655 could help glioma growth by controlling miR-214/PI3K signaling, which could be a new way to treat gliomas [89] (Figure 3 and Table 2).

Circ-PIP5K1A expression was increased in glioma tissues (compared to normal surrounding tissues), and overexpression was associated with glioma volume and histopathological grade. In vivo and in vitro overexpression of circ-PIP5K1A significantly increased glioma cell proliferation, invasion, and EMT while inhibiting apoptosis. Additionally, circ-PIP5K1A increased TCF12 expression and PI3K/AKT activation. Bioinformatics investigation established that circ-PIP5K1A and TCF12 shared a target, miR-515-5p, while the dual-luciferase reporter assay and RNA immunoprecipitation (RIP) experiment demonstrated that circ-PIP5K1A specifically targeted miR-515-5p, which bound the 3′-untranslated region (UTR) of TCF12. Circ-PIP5K1A is a possible prognostic marker for glioma and affects glioma evolution via the miR-515-5p-mediated TCF12/PI3K/AKT axis [90] (Figure 3 and Table 2).

Circ-SHKBP1 regulates the angiogenesis of endothelial cells exposed to U87 glioma (GECs). Circ-SHKBP1 expression, but not linear SHKBP1, was considerably increased in GECs compared to endothelial cells exposed to astrocytes (AECs). The absence of circ-SHKBP1 significantly reduced the viability, migration, and tube formation of GECs. In GECs, expression of miR-379/miR-544a was suppressed, and circ-SHKBP1 targeted miR-544a/miR-379 via the RNA-induced silencing complex (RISC). A dual-luciferase reporter assay revealed that miR-544a/miR-379 targeted forkhead box P1/P2 (FOXP1/FOXP2). FOXP1/FOXP2 expression was increased in GECs, and inhibiting FOXP1/FOXP2 decreased GEC viability, migration, and tube formation. At the transcriptional level, FOXP1/FOXP2 enhanced angiogenic factor with G patch and FHA domains 1 (AGGF1) expression. Additionally, through the PI3K/AKT and extracellular signal-regulated kinase (ERK1/2) pathways, knocking down AGGF1 decreased GECs’ survival, migration, and tube formation. Circ-SHKBP1 regulated the angiogenesis of GECs via the miR-544a/FOXP1 and miR-379/FOXP2 pathways, and these findings suggest that circ-SHKBP1 may be a useful target and method for combination therapy of gliomas [91] (Figure 3 and Table 2).

Circ-0067934 was a lot more common in GBM tissues and cancer cells than in normal tissues and cells. Kaplan–Meier survival curves revealed that patients with more hsa-circ-0067934 had better overall survival and a better chance of being disease-free. The functional tests showed that the knockdown of hsa-circ-0067934 slowed GBM cell growth, metastasis, and EMT and encouraged apoptosis. Furthermore, the mechanical analysis showed that the down-regulation of hsa-circ-0067934 had a big impact on the activation of the PI3K-AKT pathway. Circ-0067934 is overexpressed in GBM and plays a role in cancer cell growth and metastasis by upregulating the PI3K-AKT pathway. This means that hsa-circ-0067934 is likely to be an effective treatment for GBM [92] (Figure 3 and Table 2).

Circ-HIPK3 expression was increased in glioma tissues. Elevated circ-HIPK3 levels have been associated with a poor prognosis. Circ-HIPK3 enhances glioma cell proliferation and invasion, as well as tumor propagation in vivo, according to functional analysis. Additionally, circ-HIPK3 was discovered as a target of miR-654, whereas miR-654 was identified as a target of IGF2BP3. Circ-HIPK3 may enhance IGF2BP3 expression in glioma cells by interacting with miR-654. Finally, CCK8 and transwell experiments demonstrated that overexpression of IGF2BP3 could reverse the consequences of IGF2BP3 deficiency. Overall, these data indicate that circ-HIPK3 promotes glioma progression by targeting miR-654 from IGF2BP3, implying that circ-HIPK3 may be a therapeutic target for glioma [93] (Figure 3 and Table 2).

Circ-CFH expression was found to be greatly increased in glioma tissue and was found to be associated with tumor grade. Circ-CFH expression was also significantly increased in U251 and U373 glioma cell lines. Circ-CFH deficiency impairs cell proliferation and colony formation. Circ-CFH acts as a sponge for miR-149 and suppresses its action in U251 and U373 cells, as determined by luciferase experiments. AKT1 has since been determined as a direct target of the circ-CFH/miR-149 axis. Circ-CFH promotes glioma growth via miR-149 sponging and AKT1 signaling pathway regulation. The circ-CFH/miR-149/AKT1 axis has the potential to be a therapeutic target for glioblastoma [94] (Figure 3 and Table 2).

Circ-ABCC3 and SOX2 expression were significantly increased in glioblastoma tissues and cells; however, miR-770-5p expression was substantially decreased compared to control groups. Circ-ABCC3 expression was significantly increased in stage III glioblastoma tissues compared to stage I + II glioblastoma tissues, strongly linked to the tumor-node-metastasis (TNM) stage. Circ-ABCC3 deficiency decreased cell proliferation, migration, invasion, tube formation, and PI3K/AKT pathway activation in glioblastoma but promoted cell death. Additionally, circ-ABCC3 functioned as a sponge for miR-770-5p, directed against SOX2. Inhibitors of miR-770-5p attenuated the effects of circ-ABCC3 silencing on glioblastoma development, angiogenesis, and the PI3K/AKT pathway. Additionally, circ-ABCC3 silencing inhibited tumor growth in vivo. Circ-ABCC3 influenced the growth of glioblastoma via the miR-770-5p/SOX2 axis and the PI3K/AKT pathway. This discovery establishes a theoretical foundation for further investigation of circRNA-directed treatment for glioblastoma [95] (Figure 4 and Table 2).

#### 5.1.3. SOX4/PI3KCA Pathway

Circ-NT5E is controlled by ADARB2 binding to sites around circRNA-forming introns. Circ-NT5E may act as a sponge against miR-422a in developing glioblastoma tumors. Multiple pathogenic processes were regulated by circ-NT5E, including cell proliferation, migration, and invasion. Circ-NT5E interacted directly with miR-422a and reduced its activity. Additionally, it was revealed that circ-NT5E sponges other miRNAs, demonstrating tumor suppressor-like properties in glioblastoma. Taken together, these findings suggest that circRNA has a unique carcinogenic role in glioblastoma [96] (Figure 3 and Table 2).

#### 5.1.4. Notch Pathway

Circ-NFIX was the only circRNA overexpressed in glioma using five different experimental approaches. Additionally, when paired normal brain tissues were compared to tumor tissues, the Notch signaling system was significantly elevated. Circ-NFIX was found to behave like a sponge for miR-34a-5p, a miRNA that targeted NOTCH1. Both circ-NFIX downregulation and miR-34a-5p overexpression decreased cell proliferation and migration. Additionally, a miR-34a-5p inhibitor abolished si-suppressive circ-NFIX’s impact on glioma cells. Circ-NFIX and miR-34a-5p mimics induced apoptosis in cells. Additionally, circ-NFIX was shown in vivo to inhibit glioma growth via the regulation of miR-34a-5p and NOTCH1. Circ-NFIX expression was significantly increased in glioma cells. Circ-NFIX may enhance glioma growth via the Notch signaling pathway by sponging miR-34a-5p. This finding gave new information on the role of circ-NFIX in the progression of human glioma malignancy [97] (Figure 3 and Table 2).

#### 5.1.5. Wnt/β-catenin Pathway

The expression of circ-0082374 was higher in glioma tissues and cells. Circ-0082374 silencing inhibited glioma cell survival, migration, invasion, and glycolysis. miR-326 was a target of circ-0082374, and miR-326 silencing diminished circ-0082374’s inhibitory effect on glioma progression. SIRT1 was a miR-326 target, and circ-0082374 acted as a miR-326 sponge, promoting SIRT1 production. SIRT1 inhibition inhibited circ-0082374’s prognostic glioma-promoting action. In vivo, miR-326/SIRT1 knockdown decreased xenograft tumor development. Collectively, silencing circ-0082374 inhibited viability, migration, invasion, and glycolysis in glioma cells via a ceRNA mechanism involving miR-326 and SIRT1, thereby elucidating a novel pathogenic mechanism for glioma [98] (Figure 3 and Table 2).

Circ-0001730 knockdown inhibited glioblastoma cell motility and proliferation. SP1 binds to the promoter of the circ-0001730 host gene EPHB4, therefore boosting circ-0001730 expression. Circ-0001730 stimulated the Wnt/-catenin signaling pathway via the miR-326/Wnt7B axis. Circ-0001730 knockdown inhibited glioblastoma cell invasion. In glioblastoma cells, silencing circ-0001730 reversed the EMT phenotype. Circ-0001730 expression increased with increasing clinical stage in the clinical data analysis. In high-grade glioma samples, increased circ-0001730 expression associated with a lower overall survival and DFS rate. Through the miR-326/Wnt7B axis, circ-000173 increased proliferation and invasion in glioblastoma cells. Circ-0001730 deficiency inhibited glioblastoma cell growth by inducing cell cycle arrest at the G1/S transition [99] (Figure 3 and Table 2).

Circ-0043278 directly sponged miR-638 to increase HOXA9 expression in GBM, which can stimulate Wnt/-catenin signaling. Additionally, miR-638 directly targets the HOXA9 3’UTR, suppresses HOXA9 expression, and inhibits the two Wnt signaling effectors c-Myc and Cyclin D1, whereas overexpression of HOXA9 can partially counteract the effects of miR-638 in glioma. In both, U87 and U251 cells, silencing circ-0043278 decreases HOXA9 protein expression. miR-638 inhibition corrected the impairment of malignant tumor behavior caused by circ-0043278 silencing. The circ-0043278/miRNA-638/Homeobox A9 (HOXA9) axis had a critical role in the progression of GBM [100] (Figure 3 and Table 2).

Hsa-circ-0000177 was more common in glioma tissues and cell lines than in normal tissues and cell lines. Moreover, high levels of hsa-circ-0000177 were linked to a poor prognosis in glioma patients. In functional experiments, hsa-circ-0000177 knockdown dramatically slowed the growth and spread of glioma cells in the lab. Consistently, knocking down hsa-circ-0000177 had a big impact on glioma growth in the clinic. Hsa-circ-0000177 functions as a miRNA sponge for miR-638, which targets the FZD7 gene. Through the inhibition of miR-638, it was found that hsa-circ-0000177 enhanced FZD7 expression and activation of the Wnt signaling pathway, ultimately leading to progression of glioma growth. This study showed that hsa circ 0000177 stimulates glioma cell proliferation and invasion in vitro, while also increasing the rate of tumor progression in mice. This result also revealed that hsa-circ-0000177 may be a prognostic biomarker for glioma patients [101] (Figure 3 and Table 2).

Circ-MMP9 served as an oncogene, increased GBM, and facilitated GBM cell proliferation, motility, and invasion. Following that, circ-MMP9 acted as a sponge for miR-124, increasing GBM cell proliferation, motility, and invasion via miR-124 targeting. Additionally, cyclin-dependent kinase 4 (CDK4) and aurora kinase A (AURKA) were implicated in the GBM carcinogenesis produced by the circ-MMP9/miR-124 axis. Finally, the MMP9 mRNA transcript-binding eukaryotic initiation factor 4A3 (eIF4A3) increased circ-MMP9 cyclization and expression in GBM [102] (Figure 3 and Table 2).

In human glioma cell lines, circ-ZNF292 silencing decreased tube development. The circ-ZNF292 is involved in the development of human glioma tubes. cZNF292 is a significant circular oncogenic RNA required for tube formation to progress. It is discovered that silencing cZNF292 inhibits tube formation in glioma cells by reducing proliferation and cell cycle progression. The Wnt/-catenin signaling pathway and associated genes such as PRR11, Cyclin A, p-CDK2, VEGFR-1/2, p-VEGFR-1/2, and EGFR were used to arrest cell cycle progression in human glioma U87MG and U251 cells at the S/G2/M phase. The data indicate that suppressing cZNF292 is required for tube development and may be used as a therapeutic target and biomarker in glioma. Circ-ZNF292 is required for the proliferation and tube development of gliomas. Circ-ZNF292 action may be mediated by controlling the cell cycle and associated genes. However, more mechanistic investigations are required to elucidate the mechanism of action of circ-ZNF292 [103] (Figure 3 and Table 2).

Hsa-circ-0005114 was shown to interact with both hsa- miR-142-3p and hsa-miR-590-5p, which may play a role in glioma. Hsa-circ-0005114 was shown to be involved in insulin secretion, while APC was found to be linked to Wnt signaling. Hsa-circ-0005114-miR-142-3p/miR-590-5p-APC ceRNA axis could be involved in the formation and progression of gliomas. miR-142/3p/miR-590/5p and APC ceRNA axis may be viable targets for the therapy of GBM [104] (Figure 4 and Table 2).

Circ-RFX3 expression was significantly enhanced in both GBM cell lines and tumors. Circ-RFX3 was found to enhance the proliferation, invasion, and migration of GBM cells in several overexpression and knockdown tests. Circ-RFX3 demonstrated to function as a sponge for miR-587, and its function as a competitive endogenous RNA (ceRNA) in the formation of GBM is evaluated using dual-luciferase reporter gene and RNA pull-down tests. Additionally, PDIA3 has been shown to be a miR-587 downstream target and modulate the Wnt/-catenin pathway. Circ-RFX3 may function as a pro-cancer circRNA by encouraging the development of GBM and modulating the miR-587/PDIA3/-catenin axis. This work may identify a potential molecular target for GBM therapy [105] (Figure 4 and Table 2).

#### 5.1.6. IGF1R/Ras/Erk Pathway

Hsa-circ-0006168 and IGF1R were increased in human GBM cells and tissues, whilemiR-628-5p was downregulated in the same cells and tissues; blocking hsa-circ-0006168 and increasing miR-628-5p made A172 and LN229 cells less likely to grow and move, as well as less likely to form colonies and become apoptotic. In A172 and LN229 cells, inhibiting hsa- circ-0006168 and increasing miR-628-5p expression inhibited cell proliferation, migration, invasion, and expression of vimentin and Snail (mesenchymal markers), decreased colony formation, and increased E-cadherin (epithelial marker) and apoptosis rate. miR-628-5p silencing reversed the suppressive effect of hsa-circ-0006168 deficit, restored IGF1R function and inhibited miR-628-5p-mediated inhibition. Moreover, silencing hsa-circ-0006168 suppresses the formation of xenograft tumors in vivo and decreases the levels of Ras and phosphorylated Erk1/2 both in vitro and in vivo. IFG1R was a novel target of miR-628-5p that was mechanically targeted and sponged by Hsa-circ-0006168. Inhibiting miR-628-5p may abolish the in vitro function of hsa-circ-0006168 silencing. By competing with miR-628-5p and regulating the IGF1R/Ras/Erk pathway, silencing hsa-circ-0006168 may inhibit the growth and motility of GBM cells [68] (Figure 4 and Table 2).

#### 5.1.7. MMP2/VEGFA

Circ-ATXN1 and SRSF10 expression levels were substantially higher in GECs than astrocyte-associated endothelial cells (AECs). SRSF10 or circ-ATXN1 knockdown dramatically decreased the viability, migration, and tube formation of GECs, whereas SRSF10 knockdown functioned by blocking the production of circ-ATXN1. Additionally, when SRSF10 and circ-ATXN1 were knocked down simultaneously, the inhibitory effects on cell survival, migration, and tube formation in GECs were dramatically amplified compared to when SRSF10 and circ-ATXN1 were knocked down separately. miR-526b-3p expression was decreased in GECs. Circ-ATXN1 functionally targeted miR-526b-3p in an RNA-induced silencing complex. Increased miR-526b-3p expression decreased the viability, migration, and tube formation of GECs. Additionally, miR-526b-3p influenced the angiogenesis of GECs by suppressing the production of MMP2/VEGFA. The SRSF10/circ-ATXN1/miR-526b-3p axis was critical in regulating GEC angiogenesis. The discoveries above identified novel anti-angiogenic targets in glioma [106] (Figure 4 and Table 2).

Circ-ARF1 was found to have an oncogenic impact. ISL2 was overexpressed in gliomas and was associated with a bad prognosis. ISL2 regulated VEGFA expression in GSCs and increased the proliferation, invasion, and angiogenesis of hBMECs via ERK signaling mediated by VEGFA. Cir-ARF1 increased ISL2 expression in GSCs via miR-342–3p sponging. Additionally, U2AF2 coupled to and enhanced the stability and expression of cytoplasmic ARF1, whereas ISL2 encouraged U2AF2 expression, forming a feedback loop in GSCs. Both U2AF2 and circ-ARF1 were carcinogenic, overexpressed in glioma, and associated with a poor prognosis. In GSCs, a new feedback loop between U2AF2, Cir-ARF1, miR-342–3p, and ISL2 was discovered. This feedback loop boosted tumor angiogenesis and may be exploited as a biomarker for glioma diagnosis and prognosis, as well as for targeted therapy [107] (Figure 4 and Table 2).

#### 5.1.8. FOXM1

Circ-PIK3C2A expression increased the proliferation, invasion, and creation of tumor cells; deletion of circ-PIK3C2A function had the exact opposite effect on the tumor cells’ growth and development. The establishment of a subcutaneous xenograft tumor model in nude mice demonstrated that the loss of function of circ-PIK3C2A efficiently reduced tumor load in vivo and prolonged the survival duration of tumor-bearing animals. The interaction between circ-PIK3C2A/miR-877-5p and FOXM1 was validated using a luciferase reporter experiment. Circ-PIK3C2A acts as an endogenous competitive RNA by sponging miR-877-5p through specific binding sites, thereby altering FOXM1 expression. These findings collectively show that circ-PIK3C2A acts as a ceRNA via modulating the miR-877-5p/FOXM1 axis, opening up a unique avenue for future clinical intervention with glioblastoma [108] (Figure 4 and Table 2).

**Table 2 cells-11-02130-t002:** List of circRNAs responsible for the up-regulation of glioblastoma.

Sl. No.	CircRNA	Function	Signaling Pathway	Expression	Reference
1	Hsa-circ-0046701	Sponging miR-142-3p		Upregulated	[67]
2	Circ-UBAP2	Sponging miR-1205 and miR -382		Upregulated	[68]
3	Circ-PARP4	Sponging miR-125a-5p		Upregulated	[69]
4	Hsa-circ-0008344	miR-433-3p		Upregulated	[65]
5	Hsa-circ-0012129	Sponging miR -661		Upregulated	[70]
6	Circ-CDC45	Sponging miR-485-5p		Upregulated	[71]
7	Circ-0029426	Sponging miR-197		Upregulated	[72]
8	Circ-ENTPD7	Sponging miR-101-3p		Upregulated	[73]
9	Circ-FLNA	Sponging miR-1993p		Upregulated	[38]
10	Hsa-circ-0076248	Sponging miR-181a		Upregulated	[74]
11	Circ-LGMN	Sponging miR-127-3p		Upregulated	[75]
12	Circ-FOXO3	Sponging miR-138-5p/miR-432-5p		Upregulated	[76]
13	Circ-0074027	Sponging miR-518a-5p		Upregulated	[66]
14	Circ-SKA3	Sponging miR-1		Upregulated	[77]
15	Circ-0001801	Sponging miR-628-5p		Upregulated	[78]
16	Circ-FOXM1	Sponging miR-577		Upregulated	[79]
17	Circ-NF1	Sponging miR-340		Upregulated	[80]
18	Circ-MELK	Sponging miR-593		Upregulated	[81]
19	Circ-NUP98	Sponging miR-519a-3p		Upregulated	[82]
20	Circ-PITX1	Sponging miR-379-5p	Mapks pathway	Upregulated	[83,84]
21	Circ-ASAP1	Sponging miR-502-5p	Mapks pathway	Upregulated	[85]
22	Circ-MAPK4	Sponging miR-125a-3p	Mapks pathway	Upregulated	[86]
23	Circ-TTBK2	Sponging miR-217	Mapks pathway	Upregulated	[87]
24	Circ-0000215	Sponging miR-495-3p	CXCR2/PI3K/AKT pathway	Upregulated	[88]
25	Circ-0037655	Sponging miR-214	PI3K pathway	Upregulated	[89]
26	Circ-PIP5K1A	Sponging miR-515-5p	TCF12 and PI3K/AKT pathway	Upregulated	[90]
27	Cir-SHKBP1	Sponging miR-544a/miR-379	PI3K/AKT pathway	Upregulated	[91]
28	Hsa-circ-0067934		PI3K/AKT pathway	Upregulated	[92]
29	Circ-HIPK3	Sponging miR-654	IGF2/PI3K/AKT pathway	Upregulated	[93]
30	Circ-CFH	Sponging miR-149	PI3K/AKT pathway	Upregulated	[94]
31	Circ-ABCC3	Sponging miR-770-5p	PI3K/AKT pathway	Upregulated	[95]
32	Circ-NT5E	Sponging miR-422a	SOX4/PI3KCA pathway	Upregulated	[96]
33	Circ-NFIX	Sponging miR-34a-5p	Notch pathway	Upregulated	[97]
34	Circ-0082374	Sponging miR-326	Wnt/β-catenin pathway	Upregulated	[98]
35	Circ-0001730	Sponged miR-326	Wnt/β-catenin pathway	Upregulated	[99]
36	Circ-0043278	Sponged miR-638	Wnt/β-catenin pathway	Upregulated	[100]
37	Circ-0000177	Sponging miR-638	Wnt/β-catenin pathway	Upregulated	[101]
38	Circ-MMP9	Sponging miR-124	Wnt/β-catenin pathway	Upregulated	[102]
39	Circ-ZNF292		Wnt/β-catenin pathway	Upregulated	[103]
40	Hsa-circ-0005114-	Sponging miR-142-3p/miR-590-5p	Wnt/β-catenin pathway	Upregulated	[104]
41	Circ-RFX3	Sponging miR-587	Wnt/β-catenin pathway	Upregulated	[105]
42	Circ-0006168	Sponging miR-628-5p	IGF1R/Ras/Erk pathway	Upregulated	[68]
43	Circ-ATXN1	Sponging miR-526b-3p	MMP2/VEGFA	Upregulated	[106]
44	Circ-ARF1	Sponging miR-342–3p	MMP2/VEGFA	Upregulated	[107]
45	Circ-PIK3C2A	Sponging miR-877-5p	FOXM1	Upregulated	[108]

### 5.2. Circ-RNAs Responsible for the Down-Regulation of Glioblastoma

Circ-SHPRH generates a “UGA” stop codon by overlapping genetic codes, resulting in the translation of the 17 kDa SHPRH-146aa. In normal human brains, both circ-SHPRH and SHPRH-146aa are abundantly expressed, however in glioblastoma they are suppressed. Overexpression of SHPRH-146aa lowers the malignant behavior and tumorigenicity of U251 and U373 glioblastoma cells in vitro and in vivo. SHPRH-146aa functions by preventing full-length SHPRH from being degraded by the ubiquitin-proteasome. As an E3 ligase, SHPRH stabilization sequentially ubiquitinates proliferating cell nuclear antigen (PCNA), inhibiting cell proliferation and tumorigenesis. SHPRH-146aa, derived from circ-overlapping SHPRH’s genetic codes, functions as a tumor suppressor in human glioblastoma [109] (Figure 5 and Table 3).

In cancer cells, overexpression of FBXW7-185aa reduced proliferation and cell cycle acceleration, whereas knockdown increased aggressive phenotypes in vitro and in vivo. FBXW7-185aa decreased the half-life of c-Myc by inhibiting the stabilization of c-Myc produced by USP28. In glioblastoma clinical samples, circ-FBXW7 and FBXW7-185aa levels were significant than in paired tumor-adjacent tissues. Circ-FBXW7 expression was linked considerably with overall survival in glioblastoma patients. Endogenous circRNA encodes a functional protein in human cells, and circ-FBXW7 and FBXW7-185aa may be predictive of brain cancer prognosis [110] (Figure 5 and Table 3).

Circ-CDR1as has been found to bind to the p53 protein, which is made by the body’s cells. With each glioma grade, the amount of CDR1as in the brain decreases, and it is a good predictor of overall survival in glioma, especially in GBM. CDR1as does not act as a miRNA sponge, but it does keep p53 proteins stable by stopping them from being ubiquitinated. CDR1as directly interacts with the p53 DBD domain, which is important for MDM2 binding. This stops the p53/MDM2 complex from forming. In the case of DNA damage, CDR1as may keep p53 working and protect cells from DNA damage. Significantly, CDR1as slows down tumor growth in both the lab and in the clinic, but it does not work in cells where p53 is missing or mutated. Because CDR1as depletion may play a big role in promoting tumorigenesis by lowering p53 expression in glioma, this may be the case. These findings help us better understand the roles and mechanisms of action of circular RNAs in general and CDR1as in particular, and they could lead to new ways for the treatment of glioma [111] (Figure 5 and Table 3).

#### 5.2.1. PI3K/AKT Pathway

Circ-EPB41L5 expression levels were significantly decreased in glioblastoma tissues and cell lines relative to normal brain tissues and cell lines. Low circ-EPB41L5 expression was associated with a poor prognosis in glioblastoma patients, although overexpression suppressed glioma cell proliferation, clone formation, migration, and invasion abilities, whereas suppression had the opposite impact. RNA-seq analysis revealed that the host gene was circ-EPB41L5, which acted as a sponge against miR-19a, inhibiting miR-19a activity from upregulating EPB41L5 expression. Through EPB41L5, circ-EPB41L5 regulated RhoC expression and AKT phosphorylation. The study demonstrates that circ-EPB41L5/miR-19a/EPB41L5/p-AKT regulatory axis plays a significant role in glioblastoma progression, providing a fresh insight into the mechanisms underlying glioblastoma [112] (Figure 5 and Table 3).

Circ-AKT3 expression is reduced in GBM tissues compared to paired neighboring normal brain regions. By exploiting overlapping start-stop codons, circ-AKT3 encodes a 174 amino acid (aa) new protein called AKT3-174aa. Overexpression of AKT3-174aa inhibited cell proliferation, radiation resistance, and in vivo tumorigenicity in GBM cells, whereas circ-AKT3 knockdown improved the malignant characteristics of astrocytoma cells. AKT3-174aa interacts competitively with phosphorylated PDK1, inhibits AKT-thr308 phosphorylation, and acts as a negative regulator of the PI3K/AKT signaling intensity. Circular RNA expression of the AKT3 gene leads to the development of GBM tumors, and this data support the concept that restoring AKT3-174aa while reducing active AKT may give additional benefits for select GBM patients [113] (Figure 5 and Table 3).

#### 5.2.2. Wnt/β-catenin Pathway

In glioma tissues and cell lines, cir-ITCH expression was decreased. According to the receiver operating curve analysis, cir-ITCH has a pretty high diagnosis accuracy. The Kaplan–Meier analysis demonstrated that a decreasing cir-ITCH level was related to a decreased survival rate in patients with glioma. Cir-ITCH dramatically enhanced glioma cell proliferation, migration, and invasion. Following identification of the linear isomer of cir-ITCH, the ITCH gene was identified as the downstream target. Following that, RNA immunoprecipitation demonstrated unequivocally that cir-ITCH sponged miR-214, hence promoting ITCH expression. The gain and loss of function experiments revealed that cir-ITCH exerts anti-oncogenic activity through sponging miR-214 and modulating the ITCH-Wnt/-catenin pathway. These findings indicate that cir-ITCH functions as a tumor suppressor gene in glioma and may serve as a promising predictive biomarker for patients with glioma. Thus, re-expression of cir-ITCH may be a potential direction for developing a unique therapy technique [114] (Figure 5 and Table 3).

#### 5.2.3. VEGFA Pathway

Numerous differentially expressed circRNAs were identified, with circ-SMARCA5 and circ-FBXW7 being the most downregulated. Both are recognized as tumor suppressors in adult glioma and glioblastoma. Furthermore, in unsupervised hierarchical cluster analysis, patients with a poor prognosis were clustered independently from those with a favorable prognosis [42] (Figure 5 and Table 3).

In glioblastoma multiforme, circ-SMARCA5 functions as a sponge for the splicing factor serine and arginine rich splicing factor 1 (SRSF1) (GBM). After establishing physical contact between SRFS1 and circ-SMARCA5, the expression of total, pro-angiogenic (Iso8a), and anti-angiogenic (Iso8b) mRNA isoforms of vascular endothelial growth factor A (VEGFA), a known SRSF1 splicing target, was examined. The Iso8a to Iso8b ratio increased significantly in GBM biopsies compared to parenchyma controls, was negatively correlated with circ-SMARCA5 expression, and decreased significantly in U87-MG overexpressing circ-SMARCA5 compared to the negative control. Blood vascular microvessel density was negatively correlated with circ-SMARCA5 expression but favorably associated with SRSF1 expression. Kaplan–Meier survival analysis revealed that patients with GBM who had low circ-SMARCA5 expression had worse overall and progression-free survival rates than those with high circ-SMARCA5 expression. This data strongly imply that circ-SMARCA5 is an upstream regulator of the ratio of pro- to anti-angiogenic VEGFA isoforms in GBM cells and a highly potential prognostic and anti-angiogenic molecule in GBM [115] (Figure 5 and Table 3).

The glioblastoma cell line U87MG, overexpressing circ-SMARCA5, expressed a higher level of the serine and arginine-rich splicing factor 3 (SRSF3) RNA isoform including exon 4, which is ordinarily skipped in an SRSF1-dependent manner, resulting in non-productive nonsense-mediated decay (NMD) substrate. SRSF3 has been shown to interact with two additional splicing factors, polypyrimidine tract binding protein 1 (PTBP1), and polypyrimidine tract binding protein 2 (PTBP2), which promote glioma cell motility. Circ-SMARCA5 is highly downregulated in GBM, and it may work by influencing the activity of SRSF1, hence affecting the splicing and expression of SRSF3 and PTBP1. An in-depth examination of the relationship between circ-SMARCA5 and SRSF1 and its downstream network in GBM cells was carried out, with the goal of bolstering the case for circ-SMARCA5 as a GBM biomarker [116] (Figure 5 and Table 3).

#### 5.2.4. CDR1 Pathway

Circ-0001946 and CDR1 expressions were low in GBM cells, although miR-671p expression was high. Circ-0001946 inhibited the expression of miR-671p, hence increasing the expression of CDR1 in the miR-671p target gene. Circ-0001946 and CDR1 inhibited proliferation, migration, and invasion of GBM cells while increasing apoptosis, but miR-671p had the reverse impact. Circ-0001946 decreased GBM growth and Ki67 expression in GBM cells, as demonstrated by the xenograft mouse model and immunohistochemistry data [117] (Figure 5 and Table 3).

#### 5.2.5. WWOX Signaling Pathway

Circ-MTO1 expression was significantly decreased in glioblastoma tumors compared to neighboring normal tissues. In glioblastoma, a lower circ-MTO1 level was substantially related to shorter overall survival. circ-MTO1 decreased U251 cell proliferation. CircumMTO1 induces the expression of WWOX in U251 cells, and WWOX is involved in the circ-MTO1-induced reduction of U251 cell proliferation. miR-92 inhibits WWOX production by specifically targeting its mRNA 3’ UTR. More crucially, circ-MTO1 interacts directly with miR-92 and acts as a miRNA sponge, increasing WWOX expression. Circ-MTO1 suppresses glioblastoma cell growth via the miR-92/WWOX signaling pathway [64] (Figure 5 and Table 3).

#### 5.2.6. SMAD6 Signaling Pathway

Circ-CD44 expression was decreased in glioblastoma multiforme (GBM) tissues and primary GBM cells. LRRC4 interacts with SAM68. By blocking the interaction of SAM68 and CD44 pre-mRNA, LRRC4 enhanced the production of circ-CD44. In vivo, re-expression of circ-CD44 dramatically inhibited GBM cell proliferation, colony formation, and invasion, as well as tumor growth. Circ-CD44 acts as a ceRNA for miR-326 and miR-330-5p in GBM growth, enhancing SMAD6 expression and regulating the TGF-b signaling pathway. Thus, the LRRC4/Sam68/circ-CD44/miR-326/miR-330-5p/SMAD6 signaling axis may provide a therapeutic target for GBM [118] (Figure 5 and Table 3).

**Table 3 cells-11-02130-t003:** List of circRNAs responsible for the down-regulation of glioblastoma.

Sl. No.	CircRNA	Function	Signaling Pathway	Expression	Reference
1	Circ-SHPRH			Downregulated	[109]
2	Circ-FBXW7			Downregulated	[110]
3	Circ- CDR1			Downregulated	[111]
4	Circ-EPB41L5	Sponging miR -19a	PI3K/AKT pathway	Downregulated	[112]
5	Circ-AKT3		PI3K/AKT pathway	Downregulated	[113]
6	Circ-ITCH	Sponging miR -214	Wnt pathway	Downregulated	[114]
7	Circ-SMARCA5		VEGFA pathway	Downregulated	[115,116]
8	Circ- 0001946	Sponging miR-671-5p	CDR1 pathway	Downregulated	[117]
9	Circ-MTO1	Sponging miR-92	WWOX signaling pathway	Downregulated	[64]
10	Circ-CD44	Sponging miR-326/miR-330-5p	SMAD6 signaling pathway	Downregulated	[118]

## 6. Role of circRNAs in Other Brain Tumors

### 6.1. Pituitary Adenoma

Circ-VPS13C was shown to be considerably higher in non-functioning pituitary adenoma (NFPA) samples and cell line studies. In vitro and in vivo experiments including gain- and loss-of-function mutations revealed that silencing circ-VPS13C decreases the growth of pituitary tumor cells in vitro and in vivo. Silencing circ-VPS13C enhanced the expression of IFITM1 and activated its downstream genes involved in the MAPK- and apoptosis-associated signaling pathways. VPS13C reduced IFITM1 expression, detected by a new method, primarily by competitively interacting with RRBP1, an endoplasmic reticulum membrane ribosome-binding protein, and thereby decreasing the stability of IFITM1 mRNA. By controlling mRNA stability by interacting with ribosome-binding proteins on the endoplasmic reticulum membrane, circ-VPS13C has been demonstrated to be a key regulator in the proliferation and development of NFPAs [119].

Circ-NFIX was significantly expressed in invasive pituitary adenomas. Circ-NFIX regulated the expression of miR-34a-5p and CCNB1 during the in vivo and in vitro development and progression of pituitary adenomas. However, miR-34a-5p expression was nearly the opposite. Circ-NFIX silencing or miR-34a-5p overexpression decreased invasion, migration, and proliferation of pituitary adenoma tumors by regulating CCNB1 [120].

Overexpression of circ-OMA1 (hsa circRNA 0002316) was shown to sponge miR-145-5p, whose inhibition on NFPA cells was abolished. miR-145-5p was dramatically lowered in NFPA samples and linked adversely with NFPA invasiveness. miR-145-5p overexpression decreased NFPA cell growth and invasiveness and induced apoptosis. Translationally controlled tumor protein (TPT1) was identified as a target of miR-145-5p and as a mediator of miR-145-5p. TPT1 and its downstream components Mcl-1 and Bcl-xL were downregulated, and Bax was increased by miR-145-5p. Circ-OMA1 promotes NFPA growth by functioning as the sponge of tumor suppressor miR-145-5p to regulate the TPT1 signaling pathway, indicating a therapeutic target in suppressing the tumorigenesis of NFPAs [121].

### 6.2. Ependymoma

CircRNAs exhibits diverse expression patterns in ependymomas in comparison to controls and between patients who survived and those who had a poor prognosis, which suggests that circRNAs might be utilized as a diagnostic and prognostic biomarker in the future. CircRNAs such as circ-VCAN, circ-RMST, circ-LRBA, circ-WDR78, circ-DRC1, and circ-BBS9 were specifically upregulated in ependymomas. CircRNAs such as circ-SMARCA5, circ-FBXW7, circ-RIMS1, circ-RIMS2, and circ-EPB41L5 showed to be significantly downregulated in ependymomas [42].

## 7. Future Prospective of Circular RNAs in Brain Tumors

In this review paper, we have tried to talk about different miRNAs that are either directly or indirectly controlling the growth of glioblastoma. CircRNAs have a lot of power in the field of clinical prognosis and diagnosis because they sponge the different miRNAs and control more than one molecular pathway. The expression or function of circRNAs could be used as a possible treatment for brain tumors. This could lead to better survival rates and allow for more research into the treatment of brain tumors. There is a lot of work that needs to be done to find out how these things work, and more research needs to be done on how circRNAs work. This study, on the other hand, could help improve future molecular therapies for medulloblastoma and glioblastoma. When it comes to brain tumors biomarkers, circRNAs could be very useful in the future. Because some circRNAs are oncogenes and others are tumor suppressors, they could be used for treatment and act as brain tumors biomarkers in the future.

## 8. Conclusions

While circRNAs are gaining prominence in the research community, medulloblastoma and glioblastoma are the most prevalent CNS tumors in children and adults, respectively. As a result, the relationship between circRNAs in medulloblastoma and glioblastoma clinical features is complex and mostly unknown, and the processes causing glioma and medulloblastoma-specific circRNA expression patterns warrant additional exploration. CircRNA knockout has remained a challenge thus far. As the bulk of circRNAs are produced from genes with the ability to code, silencing or downregulating the corresponding circRNAs will definitely influence the host genes. Additionally, ubiquitous low levels of circRNA expression may inhibit their protein translation in MB, GBM, and other human disorders.

These RNA molecules might influence various biological processes associated with the spread of GBM, including cell proliferation, apoptosis, invasion, and treatment resistance. These effects are accomplished through the activation of important molecular processes and signaling pathways. Certain circRNAs have been shown to positively influence critical pathways such as the PI3K/Akt/mTOR signaling, the Wnt/catenin pathway, and the MAPK cascade, while others have been shown to negatively affect them. The advancement of recently discovered circRNA-identification methods is enhancing the diagnostic and treatment approaches for MB and GBM based on circRNAs, which will be an important moment in eradicating these life-threatening tumors. In the future, critical phases of glioma development must be researched further to identify particular circRNAs, create a comprehensive gene–protein interaction network, and clarify their role and fundamental mechanism of action by a combination of in vitro and in vivo and clinical studies. CircRNAs are predicted to become a focal point of research in the field of non-coding RNAs in the near future. Additional research will be required to confirm that circRNAs play a critical role in medulloblastoma, glioblastoma, and other disorders, hence opening up new avenues for early detection, pathological grading, targeted therapy, and prognosis of these diseases.

## Figures and Tables

**Figure 1 cells-11-02130-f001:**
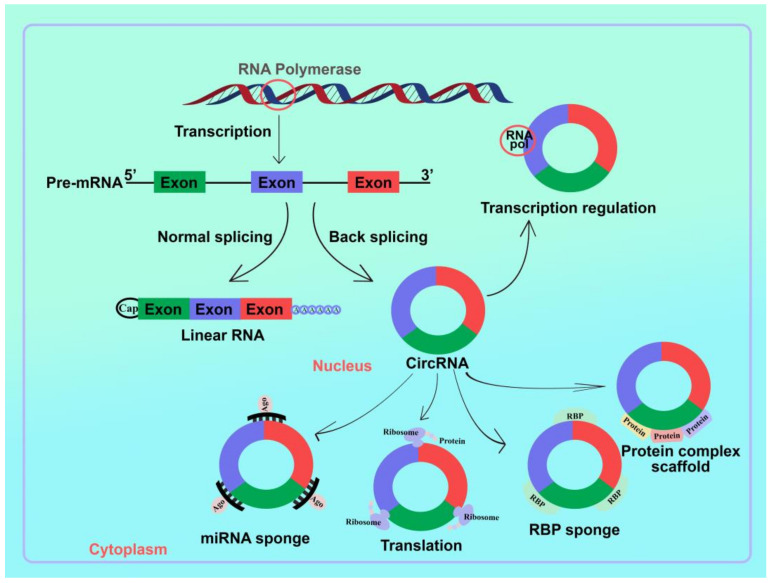
The function of circRNAs’; miRNA molecular sponging: circRNAs with miRNA binding sites can prevent miRNA from attaching to its target mRNA and hence prevent miRNA from inhibiting the target protein. Regulation of translation: circRNAs to bind to the ribosome in order to regulate translation. RBP sponging: circRNAs with an RBP binding site can regulate protein activity. Protein complex scaffold: circRNAs may serve as protein scaffolds.

**Figure 2 cells-11-02130-f002:**
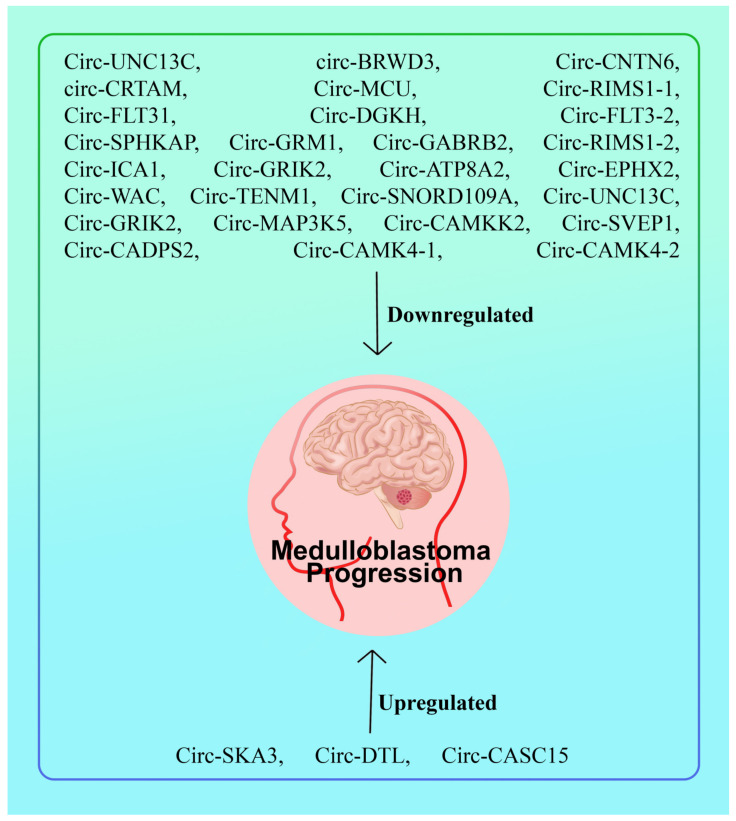
CircRNA regulation on medulloblastoma: various circRNAs including circ-UNC13C, circ-BRWD3, circ-CNTN6, circ-CRTAM, circ-MCU, circ-RIMS1-1, circ-FLT31, circ-DGKH, circ-FLT3-2, circ-SPHKAP, circ-GRM1, circ-GABRB2, circ-RIMS1-2, circ-ICA1, circ-GRIK2, circ-ATP8A2, circ-EPHX2, circ- WAC, circ-TENM1, circ-SNORD109A, circ-UNC13C, circ-GRIK2, circ-MAP3K5, circ-CAMKK2, circ-SVEP1, circ-CADPS2, circ-CAMK4-1, and circ-CAMK4-2 are responsible for the downregulation of medulloblastoma growth. Although, various circRNAs including circ-SKA3, circ-DTL, and circ-CASC15 are responsible for the up-regulation of medulloblastoma growth.

**Figure 3 cells-11-02130-f003:**
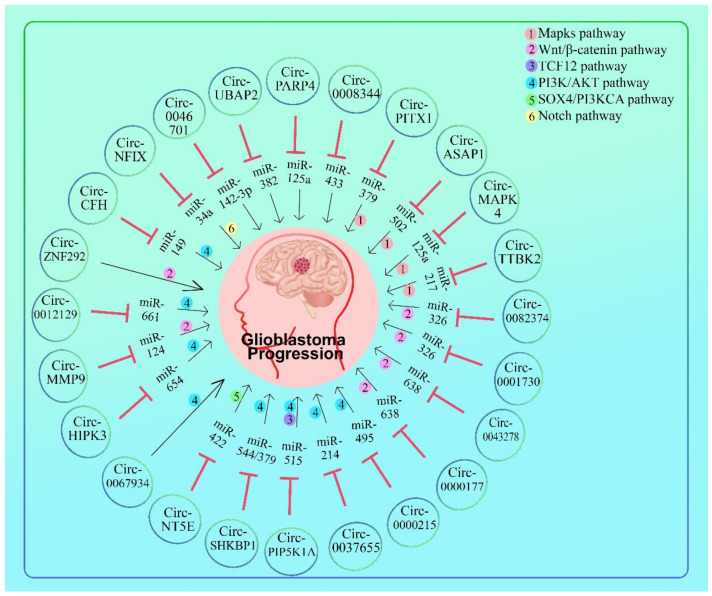
CircRNAs responsible for upregulation of glioblastoma growth: A schematic representation of the circRNAs involved in the regulation of glioma cell proliferation, migration, and invasion, among other functions. Numerous known circRNAs operate as miRNA sponges, subsequently increasing the amount of expression of the appropriate target genes. These target genes or proteins further influence downstream factors involved in cancer signaling pathways by functioning as transcription factors or regulatory proteins, as well as through other mechanisms. CircRNAs such as hsa-circ-0046701, circ-UBAP2, circ-PARP4, Hsa-circ-0008344, circ-PITX1, circ-ASAP1, circ-MAPK4, circ-TTBK2, circ-0082374, circ- 0001730, circ-0043278, circ-0000177, circ-0000215, circ-0037655, circ-PIP5K1A, circ-SHKBP1, circ-NT5E, hsa-circ-0067934, circ-HIPK3, circ-MMP9, hsa-circ-0012129, circ-ZNF292, circ-CFH, and circ-NFIX are responsible for the up-regulation of glioblastoma growth through different pathways including those of the Mapks, Wnt/β-catenin, TCF12, PI3K/AKT, SOX4/PI3KCA, and Notch. For example, circ-TTBK2 activates the Mapks signaling pathway through miR-217.

**Figure 4 cells-11-02130-f004:**
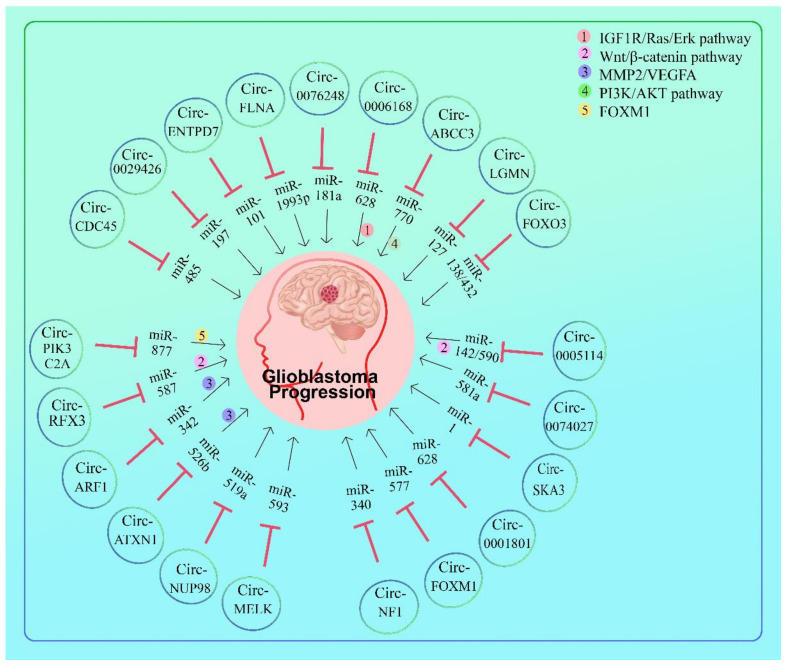
CircRNAs responsible for upregulation of glioblastoma growth: A schematic representation of the circRNAs involved in the regulation of glioma cell proliferation, migration, and invasion, among other functions. Numerous known circRNAs operate as miRNA sponges, subsequently increasing the amount of expression of the appropriate target gene. These target genes or proteins further influence downstream factors involved in cancer signaling pathways by functioning as transcription factors or regulatory proteins, as well as through other mechanisms. CircRNAs such as circ-CDC45, circ-0029426, circ-ENTPD7, circ-FLNA, hsa-circ-0076248, circ-0006168, circ-ABCC3, circ-LGMN, circ-FOXO3, hsa-circ-0005114, circ-0074027, circ-SKA3, circ-0001801, Circ-FOXM1, circ-NF1, circ-MELK, circ-NUP98, circ-ATXN1, circ-ARF1, circ-RFX3, and circ-PIK3C2A are responsible for the up-regulation of glioblastoma growth through different pathways including those of the IGF1R/Ras/Erk, Wnt/β-catenin, MMP2/VEGFA, PI3K/AKT, and FOXM1. For example, circ-RFX3 activates the Wnt/β-catenin pathway through miR-587.

**Figure 5 cells-11-02130-f005:**
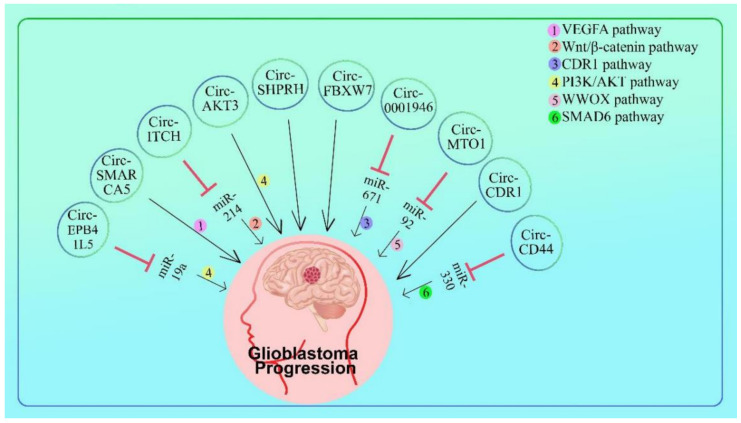
CircRNAs that downregulate glioblastoma growth: A schematic representation of the circRNAs involved in the regulation of glioma cell proliferation, migration, and invasion, among other functions. Numerous discovered circRNAs operate as miRNA sponges, therefore downregulating the amount of expression of the relevant target gene. These target genes or proteins further influence downstream factors involved in cancer signaling pathways by functioning as transcription factors, regulatory proteins. CircRNAs such as circ-CDC45, circ- 0029426, circ-ENTPD7, circ-FLNA, hsa-circ- 0076248, circ-0006168, circ-ABCC3, circ-LGMN, Circ-FOXO3, Hsa-circ- 0005114, Circ- 0074027, Circ-SKA3, Circ-0001801, Circ-FOXM1, Circ-NF1, Circ-MELK, Circ-NUP98, circ-ATXN1, circ-ARF1, circ-RFX3, and circ-PIK3C2A are responsible for the down-regulation of glioblastoma growth through different pathways including those of the VEGFA, Wnt/β-catenin, CDR1, PI3K/AKT, WWOX, and SMAD6 pathways. For example, cir-ITCH promotes the Wnt/-catenin signaling pathway via miR-214.

**Table 1 cells-11-02130-t001:** List of circRNAs responsible for the down-regulation and up-regulation of medulloblastoma.

Sl. No.	CircRNA	Function	Expression	Reference
1	Circ-SKA3	Sponging miR-383-5p/miR-326	Upregulated	[19,20]
2	Circ-CASC15		Upregulated	[21]
3	Circ-DTL		Upregulated	[21]
4	Circ-UNC13C		Downregulated	[21]
5	Circ-BRWD3		Downregulated	[21]
6	Circ- CNTN6		Downregulated	[21]
7	Circ- CRTAM		Downregulated	[21]
8	Circ-MCU		Downregulated	[21]
9	Circ-RIMS1-1		Downregulated	[21]
10	Circ-FLT31		Downregulated	[21]
11	Circ-DGKH		Downregulated	[21]
12	Circ-FLT3-2		Downregulated	[21]
13	Circ-SPHKAP		Downregulated	[21]
14	Circ-GRM1		Downregulated	[21]
15	Circ-GABRB2		Downregulated	[21]
16	Circ-RIMS1-2		Downregulated	[21]
17	Circ-ICA1		Downregulated	[21]
18	Circ-GRIK2		Downregulated	[21]
19	Circ-ATP8A2		Downregulated	[21]
20	Circ-EPHX2		Downregulated	[21]
21	Circ- WAC		Downregulated	[21]
22	Circ-TENM1		Downregulated	[21]
23	Circ-SNORD109A		Downregulated	[21]
24	Circ-UNC13C		Downregulated	[21]
25	Circ-GRIK2		Downregulated	[21]
26	Circ-MAP3K5		Downregulated	[21]
27	Circ-CAMKK2		Downregulated	[21]
28	Circ-SVEP1		Downregulated	[21]
29	Circ-CADPS2		Downregulated	[21]
30	Circ-CAMK4-1		Downregulated	[21]
31	Circ-CAMK4-2		Downregulated	[21]

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
