# Peer review of "Role of Circular RNA in Brain Tumor Development"

_cells, 2022, doi:10.3390/cells11142130_

Round 1
Reviewer 1 Report
In this manuscript, the authors review the current progress of circRNAs in the pathogenesis of MB and GBM and its underlying mechanisms and the potential targets for chemotherapy. It provides a comprehensive knowledge of circRNAs involvement in central nervous system tumors. The revised version is better.
Author Response
Thank you for correcting us.
Reviewer 2 Report
A lot of review articles have reviewed the roles of circRNA in glioma [(Dazhao Peng, et al. CircRNA: An emerging star in the progression of glioma. Biomed Pharmacother. 2022; 151:113150.) (Raziyeh Salami, et al. Circular RNAs and glioblastoma multiforme: focus on molecular mechanisms. Cell Commun Signal. 2022;20(1):13.)] and medulloblastoma [Ying-Nan Zhao, The mechanism of non-coding RNAs in medulloblastoma. Oncol Lett. 2021;22(5):758.] Here, the author submitted a manuscript to discuss the role of circular RNA in brain tumour development. However, the authors only reviewed the findings of circRNA in glioma and medulloblastoma. I suggested to add the researches of circRNA in other brain tumors such as ependymomas, pituitary tumor, pineal gland tumor et al to strengthen the novelty. However, the authors didn’t address my concerns at al. Actually, studies have found that circRNAs play important role in ependymomas [Ulvi Ahmadov et al, Distinct circular RNA expression profiles in pediatric ependymomas, Brain Pathol, 2021 Mar;31(2):387-392.] and a lot of papers have proved circRNAs are essential in pituitary tumor. [(Weiyu Zhang, CircVPS13C promotes pituitary adenoma growth by decreasing the stability of IFITM1 mRNA via interacting with RRBP1. Oncogene, 2022; 41(11):1550-1562.) (Jianhua Cheng, CircNFIX promotes progression of pituitary adenoma via CCNB1 by sponging miR-34a -5p. Mol Cell Endocrinol. 2021; 525:111140.)]. Authors should add at least one chapter discussing the role of circRNA in other brain tumors.
Author Response
Thank you. We have added the role of circRNA in other brain tumors chapter.
Reviewer 3 Report
This review discusses the circular RNA in both medulloblastoma (MB) and glioblastoma multiforme (GBM). However, several parts need to be rearranged.
1. The full name of MB should be mentioned when it appeared in the first time in the abstract and introduction.
2. In the first part of introduction, the separated description of MB and GBM were suggested. In addition, the background of MB and GBM was repeated in the 『1. Introduction』and 『2. Challenges』. The authors should rearrange the description.
3. Since the abbreviation of Circular RNA (circRNA) had been appeared once, it should be used throughout the manuscript. The repeating definitions were not needed. The authors should check the abbreviations.
4. Since Table 1 had listed all the circRNAs, the description of circRNAs in line 197 to 202 was not necessary. Additionally, Figure 2 was a repeat of Table 3. Therefore, Figure 2 should be deleted.
5. The paragraph of 5.1 should be brief and put the same phenomenon together. Subtitle was suggested.
6. All the abbreviation should be defined as it showed in the first time. The authors should check it carefully.
7. The presentation of Figure 3 and 4 should be improved. Different color block is suggested to represent different signaling pathway.
8. In the 6th section of 5.1.5., the authors mentioned “The purpose of this study is to determine the involvement of circ-ZNF292 in the creation of human glioma tubes and to determine its potential mechanism of action.” This purpose should be checked.
9. The Table 2 could be separated into down-regulation and up-regulation two different Tables.
10. The tense should be checked throughout the manuscript. Usually, present tense is applied to published papers. Moreover, the manuscript should be more concise.
Author Response
This review discusses the circular RNA in both medulloblastoma (MB) and glioblastoma multiforme (GBM). However, several parts need to be rearranged.
- The full name of MB should be mentioned when it appeared in the first time in the abstract and introduction.
Author Response: Thank you. We have inserted this correction throughout the text.
- In the first part of introduction, the separated description of MB and GBM were suggested. In addition, the background of MB and GBM was repeated in the 『1. Introduction』and 『2. Challenges』. The authors should rearrange the description.
Author Response: Thank you. In the introduction, mb and GBM are divided into two paragraphs
- Since the abbreviation of Circular RNA (circRNA) had been appeared once, it should be used throughout the manuscript. The repeating definitions were not needed. The authors should check the abbreviations.
Author Response: Thank you. We have checked the abbreviations and corrected it accordingly.
- Since Table 1 had listed all the circRNAs, the description of circRNAs in line 197 to 202 was not necessary. Additionally, Figure 2 was a repeat of Table 3. Therefore, Figure 2 should be deleted.
Author Response: Thank you. We have inserted this correction throughout the text.
- The paragraph of 5.1 should be brief and put the same phenomenon together. Subtitle was suggested.
Author Response. Thank you. We have added subtitles based on pathways. That section is too long because no further information is available about the relationship between some circRNAs and pathway.
- All the abbreviation should be defined as it showed in the first time. The authors should check it carefully.
Author Response: Thank you. We have checked the abbreviation thoroughly.
- The presentation of Figure 3 and 4 should be improved. Different color block is suggested to represent different signaling pathway.
Author Response: Thank you. We have represented different signaling pathways in different number.
- In the 6thsection of 5.1.5., the authors mentioned “The purpose of this study is to determine the involvement of circ-ZNF292 in the creation of human glioma tubes and to determine its potential mechanism of action.” This purpose should be checked.
Author Response: Thank you for this important point. This sentence has been eliminated from the text.
- The Table 2 could be separated into down-regulation and up-regulation two different Tables.
Author Response: Thank you. The table has been separated as suggested by the reviewer.
- The tense should be checked throughout the manuscript. Usually, present tense is applied to published papers. Moreover, the manuscript should be more concise.
Author Response: Thank you. We have improved the tense
Round 2
Reviewer 2 Report
The authors have mainly addressed my concerns.
Author Response
Thank you.
Reviewer 3 Report
1. The arrows in Figure 2 were askew.
2. Although the authors claimed that they have improved the Figures, there was still have space for improvement.
Author Response
1. The arrows in Figure 2 were askew.
Authors Response: Thank you for the suggestion. The arrow has been corrected as suggested by the reviewer.
2. Although the authors claimed that they have improved the Figures, there was still have space for improvement. Authors Response: Thank you for the suggestion. We have represented different signalling pathways in different colours, change the background of images and inserted border.
This manuscript is a resubmission of an earlier submission. The following is a list of the peer review reports and author responses from that submission.
Round 1
Reviewer 1 Report
In this manuscript, the authors review the current progress of circRNAs in the pathogenesis of MB and GBM and its underlying mechanisms and the potential targets for chemotherapy. It provides a comprehensive knowledge of circRNAs involvement in central nervous system tumors.
Major concerns:
The part of “the role of circRNA in glioblastoma”. As it involves many circRNAs and miRNAs and signaling pathways, the current writing makes readers boring and hard to remember anything. It will be much better if the authors write it as which circRNAs regulate which signaling pathways through which siRNAs and the possible mechanisms with subtitle. The current writing is a little bit of mess.
Minor concern:
Pay attention to the writing of some sentences and grammar.
Reviewer 2 Report
Although finding new treatment options for MB and GBM is more than necessary and requires understanding of pathophysiological mechanisms, and circRNAs are an important topic, the manuscript in its present form will not add much value to the current literature.
In my view, a review should not only collects information, it should also nicely bring findings together, accentuate and set the findings into perspective, suggesting new concepts.
This manuscript is not well prepared in these regards and lacks an attractive and well thought concept. Further, there are numerous formal mistakes, which nurses my impression of a quick and less well cared preparation of the manuscript. I indeed think, that the „market“ is flooded with such kind of reviews being rather a repetition and loose collection of information than a reinstatement value.
In more detail:
First, the authors should think about where a detailed introduction of GBM and MB is well placed instead of being redundant. In the submitted version, the diseases are described in the introduction, in chapter 2, 4 and 5 with a great degree of redundancy.
The circRNAs are suggested by authors as therapy targets. To do so, it needs to be discussed:
- how they can be specifically delivered to the tumour
- what caused their dysregulation in tumour cells
- is their dysregulated expression indeed a cause or a bystander of other genetic or epigenetic alterations? Where are they standing in the „hierarchy?“ Although downstream networks are described, the „upstream“ regulation/dysregulation is likewise important
The core paragraphs 4 and 5 are poor conceptualised. It is a confusing and loose addition of information, which requires a way better structure, logical order, in part better explanation and accentuation to be an added value to the scientific community.
Chapter 4 starts with a redundant intro about MB, suddenly, in line 174, a sentence about circRNAs is added, which is completely detached from the info given before. I would start this paragraph with the content described in line 177ff.
Then, the sentence in 190ff comes out of the blue? WHY and in which study these eight circRNAs were tested. next sentence: „Adiitionally, it Inhibited…“. What means „it“?
The whole 4th chapter requires strong revision, and the authors need to clarify what they mean with „another technique in line 221. Information is provided in a very confusing manner.
The fifth chapter likewise starts with a redundant part, and is again (albeit comprehensive) loose collection of information. This chapter needs to be split in sub-chapters. Already in line 252, the authors state that circRNAs could be drivers of tumours or tumour suppressors. this could be a separation topic for subchapters, but even more innovative „topic collections“ could be done than presenting a huge paragraph that described numerous circRNAs and what is known about them. A certain „classification“ is necessary for an added value.
As mentioned above, more information about how are these circRNAs are regulated themselves, would be necessary in my opinion. Further, I would go more in detail for certain circRNAs, and leave others just mentioned in the table. If certain „axis“ are described, than the known function of every axis-member needs to be stated.
EMT was used (line 370) before being introduced (line 400).
line 317-322 is mainly the same as the next paragraph
often „it“ is written with capital I within the sentence (eg. line347)
IDH1 wild-type glioblastoma (line 290) is not introduced
line 276: using „upregulated“ in the context of promote as in this line, does not help understanding
Apart from a clear and enormously improved concept, comprehensive spelling correction is required.
Reviewer 3 Report
Glioblastoma multiforme is among the most incurable cancers, suggesting the need for novel molecular therapies. Circular RNAs are more enriched in neuronal tissues and the aberrant expression of circular RNA implicated in the development of glioma. Although some articles have reviewed functions and clinical significance of circular RNAs in glioma. Ahmed et al systematically reviews the role of circular RNAs in brain tumor. Therefore, the review has its merit. However, this review is not exhaustive. I acknowledge that glioma and medulloblastoma are the most common types of tumors in brain. However, there are some other primary brain tumors including ependymomas, pituitary tumors, pineal gland tumors et al. Please add the findings in these tumors to strengthen the paper. In page 7, what does the GB mean? Is it a typo? It would be better if the authors review and prospect the study about targeting circular RNA for brain tumor treatment.